# Sarcoma Botryoides: Optimal Therapeutic Management and Prognosis of an Unfavorable Malignant Neoplasm of Female Children

**DOI:** 10.3390/diagnostics13050924

**Published:** 2023-03-01

**Authors:** Chrysoula Margioula-Siarkou, Stamatios Petousis, Aristarchos Almperis, Georgia Margioula-Siarkou, Antonio Simone Laganà, Maria Kourti, Alexios Papanikolaou, Konstantinos Dinas

**Affiliations:** 1Gynaecologic Oncology Unit, 2nd Department of Obstetrics and Gynaecology, Aristotle University of Thessaloniki, Hippokration General Hospital, 54642 Thessaloniki, Greece; 2Unit of Gynecologic Oncology, ARNAS “Civico–Di Cristina–Benfratelli”, Department of Health Promotion, Mother and Child Care, Internal Medicine and Medical Specialties (PROMISE), University of Palermo, 90127 Palermo, Italy; 33rd Department of Pediatrics, Aristotle University of Thessaloniki, Hippokration General Hospital, 54642 Thessaloniki, Greece

**Keywords:** sarcoma botryoides, fertility-sparing surgery, embryonal rhabdomyosarcoma, genital tract, prognosis, treatment, local debulking, neoadjuvant chemotherapy, radiation

## Abstract

Embryonal rhabdomyosarcoma (ERMS) is a rare malignancy and occurs primarily in the first two decades of life. Botryoid rhabdomyosarcoma is an aggressive subtype of ERMS that often manifests in the genital tract of female infants and children. Due to its rarity, the optimal treatment approach has been a matter of debate. We conducted a search in the PubMed database and supplemented it with a manual search to retrieve additional papers eligible for inclusion. We retrieved 13 case reports and case series, from which we summarized that the current trend is to approach each patient with a personalized treatment plan. This consists of a combination of local debulking surgery and adjuvant or neoadjuvant chemotherapy (NACT). Effort is made in every approach to avoid radiation for the sake of preserving fertility. Radical surgeries and radiation still have a role to play in extensive disease and in cases of relapse. Despite the rarity and aggressiveness of this tumor, disease-free survival and overall prognosis is excellent, especially when it is diagnosed early, compared with other subtypes of rhabdomyosarcoma (RMS). We conclude that the practice of a multidisciplinary approach is appropriate, with favorable outcomes; however, larger-scale studies need to be organized to have a definite consensus on optimal management.

## 1. Introduction

Rhabdomyosarcoma (RMS) is the most common soft tissue tumor of early childhood and young adulthood, accounting for 4 to 6% of all malignancies in this age group, with boys being affected 1.5 times more frequently than girls. The primary sites of origin are in the region of the head and neck (35–40%), followed by the genitourinary tract (25%) [1,2,3,4,5,6]. There are three major histologic subtypes of RMS described in the literature: embryonal, alveolar, and pleomorphic/undifferentiated, with embryonal rhabdomyosarcoma (ERMS) being the most common subtype (2/3 of genitourinary cases) [7]. This last one can be further classified into the classic subtype, the spindle cell subtype, and the botryoid subtype [8]. The botryoid subtype of ERMS is suggested to be the most common according to the literature. This specific rare type of tumor has an embryologic origin in the skeletal muscle cells and arises from the mucosal surfaces on the walls of hollow organs, such as the vagina, bladder, biliary tract, and nasopharynx of infants, or, more rarely, the uterine cervix [3]. Sarcoma botryoides most usually affects young people; however, it can also present in some rare cases in the elderly. It also seems that botryoid sarcoma arising from the vagina tends to develop in very young girls during infancy and early childhood [9,10]. Cervical and uterine tumors, on the other hand, primarily develop in older females with a peak incidence in the second decade [11]. The name botryoides originates from the ancient Greek root bórty(s), which indicates the appearance of “a bunch of grapes”. The typical presentation of the tumor is a nodular, grape-like mass protruding from the vagina, which should alarm every doctor since an early diagnosis is paramount to preventing death and preserving fertility in this delicate age. In the last decades, there has been a paradigm shift in the treatment of patients, including a multidisciplinary approach consisting of a variety of surgical procedures, radiation therapy, and systemic chemotherapy [1,12].

The main purpose of the present manuscript is to provide a comprehensive narrative review of the literature and summarize the main outcomes regarding the optimal therapeutic management and prognosis of this rare neoplasm of female childhood.

## 2. Methods

A literature search was performed in September 2022 through the PubMed, Scopus, and Web of Science databases. The main objective of the present study was to identify any type of research article reporting outcomes about therapeutic management and/or prognosis of cases diagnosed with botryoid sarcoma. The literature search was focused on the period 1990–2022. An electronic search was conducted by using the terms “botryoid sarcoma” [tiab] or botryoid rhabdomyosarcoma [tiab].

Observational cohort studies, both prospective and retrospective; case series; case reports; and narrative and systematic reviews that reported on the management and the prognosis of botryoid sarcoma were included in the present review. Studies were included irrespective of stage of disease at initial diagnosis and use of adjuvant therapy. The exclusion criteria concerned studies with incomplete data that did not permit definitive conclusions, non-English studies, and published abstracts without available full text.

The main outcomes of interest to identify in the included studies were age at diagnosis, primary location of the tumor, main symptom, size, stage, presence of metastases, treatment, status after treatment, diagnosis of relapse, treatment of relapse, follow up in months, and outcome as well as the main immunohistochemistry biomarkers used for final diagnosis.

Systematic search initially identified 221 papers potentially eligible for inclusion in the present analysis. After adjusting for inclusion and exclusion criteria, there were finally 13 case series or case reports included in the present review.

## 3. Results

### 3.1. Management

According to the literature search, female patients with a diagnosis of botryoides sarcoma are most commonly admitted to the hospital due to abnormal vaginal bleeding or a “grape-like” polypoid or prolapsing mass protruding from the vaginal introitus. In some cases, additional symptoms have also been described such as leukorrhea and malodorous discharge [13]. In addition, clinicians must be aware of characteristics of this unusual disease, especially the common sites of origin (vagina, bladder, etc.), the aggressiveness of the tumor, and the clinical manifestations, to avoid misdiagnosis and mismanagement, since benign polyps in the vagina or cervix are relatively uncommon in children. Furthermore, some authors also suggest that any polypoidal mass spotted in a child should be considered as botryoid RMS unless proven otherwise, given the fact that this kind of tumor can rightly be suspected in the majority of cases, and, thus, contributing to a more favorable management of the patient [14]. The initial workup usually includes imaging procedures, with first being ultrasound, followed by MRI of the primary site and regional lymph nodes, which is the best imaging method for RMS, given its superior ability to depict soft tissue changes. A computerized tomography (CT) scan and bone marrow biopsy can also provide assistance in assessing any metastatic manifestation from RMS [15], since the primary sites of metastasis in genitourinary RMSs are the lungs and the bone marrow [2,15]. According to the literature, a risk-specific approach to staging is recommended, based on the Intergroup Rhabdomyosarcoma Study Group (IRSG) clinical categorization method [16] and the TNM staging approach for rhabdomyosarcoma [8] in order to determine the patient’s clinical risk group; this will consequently stratify the treatment. Additionally, Borinstein et al. [17], in a recently published consensus article, included the tumor’s PAX/FOXO1 fusion status (positive/negative) in the risk stratification of patients, since the expression of this fusion gene is associated with dismal outcomes. Molecular testing (e.g., FISH, reverse transcription PCR, or next-generation sequencing) can readily identify PAX/FOXO1 fusions, and because results may impact treatment decisions, it was recommended by these sarcoma experts to test for FOXO1 fusions on all patients with alveolar or embryonal histology. For the diagnosis of botryoid-variant RMS, three crucial criteria have been proposed that must be fulfilled: a polypoid appearance of the lesion, an origin below a mucous membrane-covered surface, and the presence of a cambium layer [18]. However, the gold standard for the diagnosis is histopathology and post-surgery immunohistochemistry, although, in some cases, the diagnosis is achieved by preoperative histopathology or intraoperative frozen section [19]. The optimal management of botryoid RMS is a debatable matter for gynecologists. Until the 1970s, radical surgery with pelvic exenteration was regarded as the treatment of choice. Over the years, the Intergroup Rhabdomyosarcoma Study Group (IRSG) had an important impact on changing that practice, so that the frequency of radical surgery was progressively reduced from 100% in the first IRSG study to 13% in the fourth IRSG study [20,21,22]. Surgical treatment has evolved from radical exenterations to local surgical resection in appropriate candidates along with other treatment modalities that are offered such as multi-agent adjuvant or neoadjuvant chemotherapy with or without radiotherapy. Exenterative surgery still plays a role in treating persistent or recurrent tumors [1,23]. In recent years, the effective treatment for sarcoma botryoides has been considered local control of vaginal and cervical tumor with fertility-sparing methods such as polypectomy, conization, local excisions, and robot-assisted radical trachelectomy [24]. Cases were found in the literature in which vaginectomy and buccal mucosa vaginoplasty was implemented as local therapy for pediatric vaginal rhabdomyosarcoma in the spirit of local control for genitourinary RMS with the purpose of avoiding radiation. Due to the fact that about half of the cases of botryoid sarcoma affect the vagina, adequate local control is paramount in the treatment of these patients, as suggested by the high recurrence rates observed in these patients when treated with chemotherapy alone [25]. Given the high incidence of micrometastatic disease that leads to relapse in patients treated only with local therapy, all RMS patients (where possible) were treated with adjuvant chemotherapy. The current trend, as seen in the majority of cases presented in Table 1, is to begin with multi-agent NACT as the first step to downstage the tumor and then proceed with excision with a safe margin of 1 cm to 2 cm, followed by 6–12 cycles of adjuvant chemotherapy to limit the chance of recurrence. Standardized schemes of chemotherapy are based on protocols created by the IRSG. The most widely used regimen of chemotherapy for children and young adults is the combination of vincristine, actinomycin D, and cyclophosphamide (VAC), usually given in 6 to 12 cycles [26]. The recommendations of Borinstein et al. [11], who proposed a treatment algorithm based on the risk group and the gene fusion status of the patient with RMS, are consistent with this practice. The management of this tumor poses a great challenge since it mostly occurs at a young age, when the preservation of hormonal, sexual, and reproductive function is fundamental. This makes fertility-sparing procedures more enticing while radiation and radical excision are not routinely preferred. Nevertheless, they still play an important role and should be reserved for cases of relapse and for the treatment of gross residual disease following surgery or chemotherapy. This approach is further encouraged by the results of the studies performed by the IRSG, which stated that the 5-year survival among patients with nonmetastatic disease was not statistically different among those who underwent versus among those who did not undergo postoperative radiation therapy. These conclusions omitted the irrefutably negative effects of radiation on maintaining the fertility of the young [8]. While surgery can be considered for lesions that can be resected with minimal morbidity, radiotherapy is often used to treat the primary tumor site, if not initially treated, and to treat metastatic sites when such therapy is feasible. Furthermore, it can be observed that there are patients who complete therapy for RMS and frequently are not able to achieve complete radiographic response by cross-sectional imaging, although their PET scans are often normal. This finding is likely due to tumor scarring or differentiation. At this point, it is suggested that resection or biopsy of a residual tumor is not recommended except for the cases in which they are enlarging or causing pain, because the extent of the tumor response does not predict survival. These cases that remain PET avid are challenging to manage, and the decision whether to biopsy or resect a residual PET-avid tumor must be made on a case-by-case basis, weighing the risks of morbidity versus the benefit.

### 3.2. Prognosis

As with most malignancies, the key prognostic factor for the prognosis of botryoid RMS is the extent of disease and early disease stage at diagnosis. Overall, soft tissue sarcomas tend to have a dismal prognosis with a high recurrence risk for all stages, ranging from 45 to 73% (40% recurrence in the lung, 13% in the pelvic area). Furthermore, a great number of patients present with recurrence within the first 2 years after primary therapy [37,38]. However, despite its malignancy and rarity, botryoid sarcoma is associated with a very favorable prognosis (95% survival at 5 years), which has seen a dramatic improvement in recent years through the utilization of multidisciplinary treatment [12]. As indicated by the included studies, the majority of cases do not present distal metastases, which also attributes to the favorable outcomes. Several studies over the years indicated the dramatically improved prognosis, with Raney RB Jr et al. [39] highlighting the 5-year overall survival rates of 87% in patients with early-stage disease and Raney RB et al. [40] reporting overall survival rates up to 97%. Hawkins DS et al. [41] demonstrated that lesions arising from the cervix, which are more common among children than among adult patients, appear to have a better prognosis than the ones arising from other parts of the female genital tract. Nonetheless, although patients with recurrent RMS tend to have poor long-term prognosis, the 5-year survival rate after recurrence for botryoid ERMS versus other embryonal tumors is more favorable, reaching 64% vs. 26%, respectively [42]. It is obvious that long-term follow up is necessary to guarantee adequate oncological and functional results.

## 4. Discussion

RMS is a rare tumor in childhood and adolescence, accounting for 4–6% of pediatric cancers. The female genital tract is considered the prognostically favorable site, given the improved outcomes during the last several decades. Botryoid sarcoma accounts for the majority of cases of the most common RMS histologic subtype: embryonal RMS. It is found under the mucosal surface of body orifices such as the vagina, bladder, and cervix and accounts for around 10% of all RMS cases. Until today, no clear risk factor for botryoid sarcoma could be identified with certainty due to the low number of published cases. The vast majority of cases occur sporadically. Data from a number of literature reports mention the following risk factors: aging, a certain race (African-American women have double the incidence of White Americans), 5 or more years of tamoxifen prescription, and history of radiation exposure. However, the parity, age of menarche, and menopause were not identified to affect the occurrence of RMS [43]. Chemical exposure, maternal age greater than 30 years, low socioeconomic position, and environmental factors all led to the development of RMS, according to one study [44]. 

### 4.1. Molecular Pathways Involved in Botryoid Sarcoma

The pathophysiology behind the formation of sarcoma botryoides remains unknown until today. The greater percentage of children who present with this malignancy have no antecedent risk factors. However, it is more likely to develop in individuals with familial diseases that induce mutations in genes responsible for cell proliferation and death (such as Li–Fraumeni syndrome). Despite the mainly sporadic character of the malignancy, a small portion of cases have been associated with genetic diseases such Li–Fraumeni cancer susceptibility syndrome, familial pleuropulmonary tumor, neurofibromatosis Type I, and Beckwith–Wiedemann syndrome. However, the incidence may be higher in patients diagnosed with RMS before the age of 3 [3,45]. Specific gene alterations such as KRAS activation and p53 inactivation have been linked to the presentation of RMS. Most embryonal rhabdomyosarcomas, in particular, have a point mutation in exon 6 of the p53 gene on chromosome 17. In a family, the heterozygous p53 germline mutation was reported as the source of the Li–Fraumeni cancer susceptibility syndrome, which presents as a cluster of soft tissue cancers (including sarcomas). Dehner et al. [46] also reported a connection between the blastoma family and pleuropulmonary tumors, as well as confirming DICER1 autosuggest, implying that RMS in children must be treated in a broader context to account for the possibility of pleuropulmonary blastoma familial tumor predisposition syndrome [47,48,49]. The identification of DICER1 mutation is notable since it is found in 60% of Sertoli–Leydig cancers, and germline mutations found in Dicer1 increase the possibility of developing rare cancers. Mousavi and Akhavan [14] revealed the occurrence of cervical sarcoma botryoides in two sisters, suggesting that hereditary factors may play a role in the development of sarcoma botryoides [50,51,52]. Malignant mixed Mullerian tumor, widely known as carcinosarcoma, can develop exophytically from the uterine wall or cervix and have a sarcomatous gross and microscopic appearance. Nevertheless, malignant mixed Mullerian tumors usually tend to affect older people, in contrast with ERMS [53]. For the time being, it seems reasonable at a minimum to strongly consider referral to genetic counseling for patients who are younger at diagnosis, whose tumors have anaplastic features, or who have a significant family history of malignancy.

### 4.2. Diagnostic Approach

According to the literature, the diagnosis of this tumor is difficult to make, but as far as the management of this tumor is concerned, there are a variety of approaches in the treatment armamentarium, ranging from extreme, radical procedures to more conservative ones. Nuclear MRI is the gold standard for determining where the tumor originates from (whether it is in the endometrium, myometrium, or cervix) as well as the spread and involvement of neighboring structures. Because of its rarity and high-risk, malignant nature originating in the embryonic mesenchyme, botryoid sarcoma should be suspected in young-age females with vaginal bleeding or a prolapsed mass, since typically the tumor develops behind the mucosal membrane of the organs, forcing the growth to take on a characteristic grape-like form. The importance of histology in RMS prognosis cannot be overstated. Although there are three types of RMS (embryonal, alveolar, and undifferentiated), the embryonal type is the more prevalent and has a better prognosis than the alveolar type, which is rare and has a worse prognosis [18]. The Intergroup Rhabdomyosarcoma Studies (IRS) classifies RMS based on (i) the main site, (ii) tumor size, (iii) lymph node involvement, (iv) surrounding tissue infiltration, and (v) the occurrence of metastases [54]. The stage is established using two systems: the Intergroup Rhabdomyosarcoma Study Group clinical categorization method (Table 2) [16] and the TNM staging approach for rhabdomyosarcoma (Table 3) [8]. Table 2 reports the Intergroup Rhabdomyosarcoma Study Group clinical classification system for rhabdomyosarcoma. It is actually a classification of rhabdomyosarcoma cases into four clinical groups, based on the extent of disease, resectability, and margin status.

### 4.3. Therapeutic Modalities

It is of paramount importance to organize a personalized treatment plan, considering the extent of the disease and the fertility preservation. The management of botryoid rhabdomyosarcoma poses a great challenge for gynecologists. In the past, the traditional treatment for these types of tumors involved exenterative procedures, but today, modalities such as fertility-sparing methods, e.g., polypectomy, conization, local excisions, and robot-assisted radical trachelectomy, are offered and are the ones mostly implemented for the preservation of the reproductive ability. In the last decades, a variety of procedures have been added to the options of pediatric genitourinary and anorectal reconstruction. Buccal mucosa grafts are now widely employed in both adult and pediatric urology for urethral reconstruction with acceptable results. Recent reports have established that buccal mucosa vaginoplasty leads to good outcomes in patients with Mayer–Rokitansky–Kuster–Hauser syndrome (MRKH—agenesis of the Mullerian structures and vagina), complete androgen insensitivity syndrome, and repair of urogenital sinus, as far as cosmetic and functionality results are concerned. Evidence suggests that the grafts retain favorable characteristics over time and adapt well to rapid growth demands such as those imposed by puberty. Harvesting the graft is straightforward and associated with minimal morbidity. In addition, buccal grafts greatly resemble the vaginal tissue that they are to replace. Nevertheless, as with every newly introduced procedure, long-term follow up is necessary to assess the oncological outcomes since it does not represent a standard of care but rather an intervention that holds promise as a viable option with minimal esthetic impact.

It was the first IRGS trial (1972 and 1978) that recommended systematic chemotherapy following extensive surgery such as radical hysterectomy or pelvic exenteration for ERMS of the genital tract [54]. The second IRGS trial (1978–1984) suggested NACT for the first time to minimize the extent of the tumor, allowing for a less radical surgery [39]. Multi-agent adjuvant chemotherapy with or without the addition of radiotherapy plays a substantial role in the effective management of sarcoma botryoides, apart from the surgical resection of the tumor. In clinical practice, there are standardized schemes of chemotherapy that can be used preoperatively to minimize the volume of the tumor or after surgical resection to limit the chances of recurrence. The most frequently used regimen of chemotherapy for children and young adults with nonmetastatic disease is the triplet of vincristine, actinomycin D, and cyclophosphamide (VAC), and it is based on the protocols of IRSG [26]. Unfortunately, there are several toxic effects that are associated with chemotherapy, and sometimes it is not well tolerated by the patients who undergo it. The most usual side effects of cyclophosphamide that are well documented are bone marrow suppression and subsequently susceptibility to infections, hemorrhage cystitis, cardiotoxicity, and gastrointestinal disturbances. Vincristine on the other hand promotes the production of severe neurotoxicity in patients and less commonly myelosuppression, alopecia, and SIADH.

Other regimens consist of VAC plus VAI (vincristine, actinomycin D, and ifosfamide or VIE (vincristine, ifosfamide, and etoposide) plus VAC for 12 months. A randomized controlled trial by Amdt et al. compared the VAC regimen and the combination of vincristine, topotecan, and cyclophosphamide for the treatment of moderated-risk rhabdomyosarcoma. The results suggest that topotecan was not indicated to be more efficient than actinomycin D, with 68% and 73% 4-year survival rates, respectively. Irinotecan is another drug that is currently under examination for its efficacy in the treatment of pediatric rhabdomyosarcoma when combined with the VAC regimen [55].

Surgery and/or radiation still play an important role in the management of high-risk RMS with oligometastatic disease to minimize treatment failures. Aggressive surgical local control most of the time offers the advantage of sparing these young patients radiation-associated complications. However, it should be kept in mind that surgical resection is not without its own possible complications, including wound infections, fistulas, and stenosis [56]. In case of widespread metastases at presentation, local control is often postponed until later in treatment and may be customized to focus on the most symptomatic or critical sites. In almost all of these advanced cases, palliative treatment remains the only option. Although the outcome is not always favorable for the patients, the prognosis of botryoid sarcomas has dramatically improved in recent years through the combination of chemotherapy, radiotherapy, and/or surgery. Similar to the case for most other cancers, the prognosis depends on the tumor size, the histological variant, and the depth to which the disease has spread to adjacent structures at the time of diagnosis. It appears that there is a more favorable prognosis for tumors arising from the cervix compared with the ones arising from other parts of the female genital tract. Generally, the 5-year survival rate for sarcoma botryoides is 83%, 70%, 52%, and 25% for clinical stages I–IV, respectively. Unfortunately, despite the advances in therapeutic modalities, there are several reports of tumor recurrences, with the pelvis being the most common region for primary recurrence. Surprisingly, the 5-year overall survival was equally excellent, reaching 87% in nonmetastatic tumors [57]. Consistent with these results was the publication of Brand et al. [58], in which the patients’ survival rate was 80% at 68 months with the use of multimodality therapy (conservative surgery combined with chemotherapy). The identification of nodal metastases through imaging is critically important in the treatment of RMS, and tissue sampling must be performed for all patients with clinically or radiographically suspicious lymphadenopathy. In conclusion, we believe that a combination of debulking surgery, chemotherapy, and in cases of treatment-resistant tumor or remaining disease, radiation therapy demonstrates an appropriate approach in well-selected patients with botryoid sarcoma. This approach provides excellent oncologic outcomes and a low complication rate, taking into account the tumor’s location, stage, and the patient’s overall characteristics. Nevertheless, since most of the data come from case reports, larger studies with longer follow-up must be conducted in order to determine the most effective treatment guidelines.

### 4.4. Limitations and Advantages

The present summative review is, to the best of our knowledge, the first one trying to summarize the main literature outcomes about therapeutic management and prognosis. The main limitation of the present manuscript is the fact that it is a narrative review only summarizing results of relatively low-level evidence, such as retrospective series and case reports, as no prospective RCTs or large prospective cohorts were identified through the literature search. However, despite the fact that the level of evidence is low, this is relatively reasonable because the rarity of the disease poses reasonable difficulties in the conduct of level-I evidence studies. Furthermore, as our review finally summarized the main conclusions about the therapeutic modalities and prognostic outcomes, this might be the initial step to organize prospective observational cohorts, rather than multicenter ones, in an attempt to globalize the standards of practice and, thereafter, improve the outcomes of such a demanding clinical entity.

## Figures and Tables

**Table 1 diagnostics-13-00924-t001:** Characteristics and treatment of female patients with botryoid sarcoma [18,25,27,28,29,30,31,32,33,34,35,36].

Paper	Age (Months/Years)	Entry Year	Site	Symptom	Size	Stage CG/TNM	Metastases	Treatment	STATUS after Treatment	Relapse	Treatment of Relapse	Follow Up	Immunohistochemistry
Pańczak K et al., 2017 [27]	4 months	2017	**Vagina**	Vaginal bleeding and a mass protruding from the vagina, clitoromegaly	2.5 cm × 2.3 cm × 4.3 cm			Chemotherapy (VAI x7) + vaginoscopic resection R0 + VAI x2 + (adriamycin, cyclophosphamide, carboplatin, topotecan, trofosfamide, idarubicin, vincristine, and etoposide) × 3	Recurrence of vaginal RMS, qualified for radical surgery (vaginal resection)	Yes, after 14 months	Radical surgery (vaginal resection)		Desmin, myogenin (myogenic factor 4), myogenic differentiation 1, Wilms’ tumor gene expression, and Ki- 67 protein in about 90% of cells
van Sambeeck S J et al., 2014 [28]	17 months	2014	**Vagina**	Abnormal vaginal bleeding and vaginal tissue loss with a “grape bunch” appearance	6.9 × 3.7 × 4.1 cm		No distant metastases	Chemotherapy VAI x9, radical surgery or radiotherapy was omitted		Yes, 6 months	Chemotherapy and brachytherapy	Complete remission for almost 1 year	Focal positivity for desmin and myogenin
Rodrigo L. P. Romao et al., 2017 [29]	30 months	2017	**Vagina**			Stage I/group III		VAC + subtotal vaginectomy (24 weeks) with vaginal reconstruction with buccal mucosa grafts				34 months	Fusion negative
ALSaleh N et al., 2017 [30]	18 months	2017	**Uterus, cervix, and vagina**	Vaginal bleeding (8 mo) and a mass protruding through the introitus (12 mo) + difficult mictur- ition	10 × 6 cm		Without abdominal or pel- vic lymphadenopathy	Chemo VAC x10; after remission: total abdominal hysterectomy, bilateral salpingectomy with upper vaginectomy, ureterolysis, and bilateral ovarian transposition (oopexy) + VAC x5		No		12 months disease free on remission with no complaints	
Imawan D K Et al., 2019 [31]	36 months	2019	**Cervix**	Protruding mass in vagina, with a tendency to bleed	10 cm × 10 cm			Tumor excision		Yes, after 3 months	Wide excision with a 2 cm margin of healthy tissue without intraoperative biopsy + VAC x6	18 months post chemotherapy → still in remission, alive and well 44 months after	(+) Anti- desmin and anti- myogenin antibody
May T et al., 2018 [32]	24 months	2018	**Cervix**	Mass protruding through the vaginal introitus	10 × 4.0 × 4.5 cm	Stage I, group III rhabdomyosarcoma		Vaginal portion of the mass was resected + chemo (alternating vincristine, dactinomycin, and cyclophosphamide/vincristine and irinotecan) × 2 + radical trachelectomy + chemo × 12		No		12 months disease free on remission with no complaints	
Neha B et al., 2015 [18]	14 years old	2007	**Cervix**	Mass protruding from the introitus and white discharge that was occasionally blood-stained				Radical hysterectomy + VAC x6				8 months after surgery, acquired a varicella zoster virus, died due to septic shock and multiple organ failure	
Yasmin F et al., 2015 [33]	7 months		**Cervix**	Protruding mass in the vaginal area for 7 days	9.5 × 7.4 × 10 cm^3^			Surgery (subtotal hysterectomy) and chemotherapy (5 cycles, no explanation about the regimen)		Yes (2 months after chemotherapy)	Total hysterectomy and chemotherapy × 5 (no further explanation, advised for × 14)		
Bouchard-Fortier G et al., 2016 [34]	14 years old	2016	**Cervix**	Mass protruding through the vagina accompanied by uterine bleeding	5.3 × 2.9 × 6.7 cm			Robotic-assisted radical trachelectomy + 35 of 43 weeks of VAC alternating with vincristine and irinotecan				No evidence of disease 10 months following diagnosis	ERMS with diffuse anaplastic features and heterol- ogous (cartilage) differentiation
Bouchard-Fortier G et al., 2016 [34]	20 years old	2016	**Cervix**	Heavy vaginal bleed- ing and a mass protruding through the vaginal introitus	5.9 × 3.9 × 2.9 cm			Hysteroscopy + cervical conization (after the mass had detached) + 4 cycles of VAC followed by 4 cycles of VA				No evidence of disease 25 months from diagnosis	
Bouchard-Fortier G et al., 2016 [34]	21 years old	2016	**Cervix**	One-year history of abnormal uterine bleeding	3.3 × 1.7 × 2.8 cm			6 cycles of VAC + LEEP + robotic-assisted radical trachelectomy and placement of an abdominal cerclage				No evidence of disease 21 months after diagnosis	
Bell S G et al., 2021 [35]	17 years old	2021	**Cervix**	One-year history of an enlarging mass protruding from the introitus associated with vaginal bleeding				Underwent polypectomy of the mass using electrocautery, and su- ture margins were negative. + 6× cycles of vincristine, actinomycin-D, and cyclophosphamide					
Melo A et al., 2012 [36]	20 years old	2012	**Cervix**	Postcoital vaginal bleeding over 1 year			No distant metastases	radical surgery, excision of the upper third of the vagina + adjuvant chemotherapy, consisting of 4 cycles of IVA pattern+ Mesna and further 5 cycles of vincristine and actinomycin.		No		At 3 years after diagnosis, patient remains in complete remission and without clinical signs of ovarian failure	Cell positiveness for actin, vimentin, Myo D1 and desmin
Michlitsch J G et al., 2017 [25]	11 months	2017	**Vagina**	Tumor fragments were passed per vagina		Stage 1, group IIa	No distant metastases	Partial vaginectomy, converted to a total vaginectomy + VAC therapy				38 months of follow up, patient remains disease free with no evidence of local or distant recurrence	
Michlitsch J G et al., 2017 [25]	30 months	2017	**Vagina**	Exophytic vaginal mass		Stage 2, group III tumor	No distant metastases	VAC therapy + surgical resection and reconstruction (at 24 weeks)				Disease-free at 41 months following diagnosis, with no evidence of recurrence	
Michlitsch J G et al., 2017 [25]	24 months	2017	**Vagina**	Vaginal bleeding			No distant metastases	VAC therapy + total vaginectomy with reconstruction (at week 20)				Disease-free at 43 months with no evidence of recurrence	
Michlitsch J G et al., 2017 [25]	25 months	2017	**Vagina**	Protruding vaginal mass			No distant metastases	VAC therapy + anterior vaginal resection of roughly 180-degree circumference and vaginal reconstruction				Disease-free at 16 months with no evidence of recurrence	

VAC—vincristine, actinomycin D, cyclophosphamide; VAI—vincristine, actinomycin D, ifosfamide; LEEP—loop electrosurgical excision procedure.

**Table 2 diagnostics-13-00924-t002:** Intergroup Rhabdomyosarcoma Study Group Clinical Classification System for Rhabdomyosarcoma.

Clinical Group	Extent of Disease, Resectability, and Margin Status
I	A: localized tumor, confined to site of origin, completely resected.
	B: localized tumor, infiltrating beyond site of origin, completely resected.
II	A: localized tumor, gross total resection, but with microscopic residual disease.
	B: locally extensive tumor (spread to regional lymph nodes), completely resected.
III	A: localized or locally extensive tumor, gross residual disease after biopsy only.
	B: localized or locally extensive tumor, gross residual disease after major resection (≥50% debulking).
IV	Any size primary tumor, with or without regional lymph node involvement, with distant metastases, irrespective of surgical approach to primary tumor.

**Table 3 diagnostics-13-00924-t003:** TNM Staging System for Rhabdomyosarcoma.

Stage	Sites	T	Tumor Size Designation	N	M
I	Orbit Head and neck * Genitourinary † Biliary tract	T1 or T2	a or b	Any N	M0
II	Bladder or prostate Extremity Cranial parameningeal Other ‡	T1 or T2	a	N0 or Nx	M0
III	Bladder or prostate Extremity Cranial parameningeal Other ‡	T1 or T2	a	N1	M0
IV	All	T1 or T2	a or b	N0 or N1	M1

T1, tumor confined to the anatomic site; T2, tumor extension; a, ≤5 cm in diameter; b, >5 cm in diameter; N0, nodes not clinically involved; N1, nodes clinically involved; Nx, clinical status of nodes unknown; M0, no distant metastases; M1, distant metastases present. * Excluding parameningeal sites. † Nonbladder and nonprostate. ‡ Includes trunk, retroperitoneum, etc., excluding biliary tract.

## Data Availability

No new data were created or analyzed in this study. Data sharing is not applicable to this article.

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
