# Peer review of "Sarcoma Botryoides: Optimal Therapeutic Management and Prognosis of an Unfavorable Malignant Neoplasm of Female Children"

_diagnostics, 2023, doi:10.3390/diagnostics13050924_

Round 1

Reviewer 1 Report

This research article by Margioula-Siarkou et al. 2023, titled “Sarcoma botryoides: optimal therapeutic management and prognosis of an unfavourable malignant neoplasm of female children  

Title in 15 words reflects the hypothesis, and the disease, patients, and looking for whats?.

The Abstract/Keywords Section

·                Abstract appropriately summarize the manuscript, with appropriate keywords,

·                But, need more key words,

·                No discrepancies between the abstract and the manuscript remainder,

·                The Abstract can be understood without reading the manuscript; however, the manuscript provides more clarifications and details.

The Introduction Section

·                The introduction is showing details,

·                The purpose of the study is not clearly defined, as missing to mention the computational analysis,

·                Authors provided a rationale for performing the study, but still not appropriate,

·                Authors defined terms used in the remainder of the manuscript,

·                This manuscript is “Review” but needs to specify if narrative or systematic

·                Aim needs more clarification through definite objectives.

Methods and indexes need more elaboration and references as well as the follow up rational and no exclusion criteria,

The Results Section

·                Results are clearly explained,

·                Results are reasonable,

Tables are appropriate, adequately showing results and are appropriately labeled, and legend provides a clear explanation, in the results and in the annex parts.

Table 2 is not justified or explained,

No Figures or Graphs,

Discussion concise,

·                Hypothesis was proposed, but the authors didn’t state whether the hypothesis was verified or falsified in the discussion part,  

·                Authors' conclusion(s) were justified by the results found in the study,

·                Authors didn’t note “limitations” of the study, or the “strength(s)” of the study,

·                Authors should mention “recommendation” or “future prospective” for completion of the work?

List of abbreviations are needed.

The References Section

·                More new relevant references are necessary,

·                No 2022, only few ref. 2020 ? one 2021

The final recommendation Reconsider with Minor Revision.

Reviewer 2 Report

Thanks for this interesting descriptive review and your effort. 

In the methodology: why you only select PUBMED to find your cases presented in this review, do you try in other literature seekers, such as, Web of Science? 

Can you give us more information about why the selected articles were included and how many articles were discarded and why?

For example: It is a rare disease why you only select english studies? The other languages studies did not meet with all the characteristics that you mention in your table, like treatment, survival, pathological findings? 

It could be more elegant and methodological accepted if your mention in your methodology if you mention that your review of the literature was based on your main goals such as age range, female sex, type of treatment given, survival, and pathological analysis, etc. 

There are some edition mistakes such as consensus in the line 88, and some bond words in the table 1. 

In the discussion the authors did an interesting review of the principal molecular pathways involved in this sarcoma that can be part of the principal text, as a topic or section, for example, principal molecular pathways involved in Botryoid sarcoma. It can be the same for the rest of the discussion, you can move the section of the image diagnostic, surgical treatment and chemotherapy as a topic of the main document. And you can add a section of future posible target therapies based on the molecular findings. 

In the discussion, give us more information of others efforts similar than yours and their findings. 
